# A biophysical minimal model to investigate age-related changes in CA1 pyramidal cell electrical activity

Erin C. McKiernan[1]*, Marco A. Herrera-Valdez[2]*, Diano F. Marrone[3,4]

1 Departamento de Física, Facultad de Ciencias, Universidad Nacional Autónoma de México, Ciudad de México, CDMX, México, 2 Laboratorio de Dinámica, Biofísica y Fisiología de Sistemas, Departamento de Matemáticas, Facultad de Ciencias, Universidad Nacional Autónoma de México, Ciudad de México, CDMX, México, 3 Department of Psychology, Wilfrid Laurier University, Waterloo, ON, Canada, 4 McKnight Brain Institute, University of Arizona, Tucson, AZ, United States of America

* emckiernan@ciencias.unam.mx (ECM); marcoh@ciencias.unam.mx (MAHV)

**Data Availability Statement:** Resources generated by this study (code, figures, and manuscript files) are available via GitHub (github.com/emckiernan/agingCA1) and archived via Zenodo (doi.org/10.5281/zenodo.6788229). To facilitate reuse,

## Abstract

Aging is a physiological process that is still poorly understood, especially with respect to effects on the brain. There are open questions about aging that are difficult to answer with an experimental approach. Underlying challenges include the difficulty of recording *in vivo* single cell and network activity simultaneously with submillisecond resolution, and brain compensatory mechanisms triggered by genetic, pharmacologic, or behavioral manipulations. Mathematical modeling can help address some of these questions by allowing us to fix parameters that cannot be controlled experimentally and investigate neural activity under different conditions. We present a biophysical minimal model of CA1 pyramidal cells (PCs) based on general expressions for transmembrane ion transport derived from thermodynamical principles. The model allows directly varying the contribution of ion channels by changing their number. By analyzing the dynamics of the model, we find parameter ranges that reproduce the variability in electrical activity seen in PCs. In addition, increasing the L-type $Ca^{2+}$ channel expression in the model reproduces age-related changes in electrical activity that are qualitatively and quantitatively similar to those observed in PCs from aged animals. We also make predictions about age-related changes in PC bursting activity that, to our knowledge, have not been reported previously. We conclude that the model's biophysical nature, flexibility, and computational simplicity make it a potentially powerful complement to experimental studies of aging.

## Introduction

As we age, our brains undergo many changes [1, 2], but we understand relatively little about these and their effects on neural function. What does normal neurophysiological aging look like and what are the various stages? How does the electrical activity of neurons change and what are the biophysics underlying those changes? How do aging neurons respond to input from other cells? Answering these questions is not just fundamental to understanding aging as

resources are shared under open licenses (see the license file in our GitHub repository). To promote reproducibility, Python code is embedded in a Jupyter notebook that explains the code, how to use it, and how to generate the figures herein, plus additional ones.

**Funding:** This work was supported by grants from the Natural Sciences and Engineering Research Council of Canada, as well as the Ontario Mental Health Foundation, awarded to DFM. This work was also supported by DGAPA-UNAM-PAPIIT IA209817 awarded to ECM; and by DGAPA-UNAM-PAPIIT IA208618 & IN228820, and DGAPA-UNAM-PAPIME PE114919 awarded to MAH-V. The funders had no role in study design, data collection and analysis, decision to publish, or preparation of the manuscript.

**Competing interests:** The authors have declared that no competing interests exist.

a neurophysiological process, but also to understanding how this process may be altered in age-related disorders of clinical importance such as Alzheimer's [3] and Parkinson's [4] disease.

Many aging studies have focused on the hippocampus, an area of the brain involved in learning, memory formation, and spatial processing [1, 2]. Aged rats [5–7] and humans [8] show impaired learning of hippocampal-dependent spatial tasks. Long-term potentiation (LTP), a proposed physiological substrate of memory formation, has been investigated in the hippocampus and its induction and maintenance shown to be impaired in aged rats [9, 10]. A short-term form of plasticity, frequency potentiation/facilitation (FP/FF), is also impaired in hippocampal pyramidal cells (PCs) from aged rats and correlates with learning deficits [11].

Plasticity changes and behavioral impairments may result in part from altered $Ca^{2+}$ signaling in aged neurons [1, 2]. Compared to CA1 PCs from young animals, PCs from aged animals show larger and longer post-burst afterhyperpolarizations (AHPs) [12–14]. AHPs are mediated by $Ca^{2+}$-dependent $K^+$ currents, which can act like brakes on the electrical activity of CA1 PCs [15, 16]. As a result, PCs show increased spike frequency adaptation and fire fewer action potentials (APs) in response to acute stimuli or during bursting activity [17–19]. Larger AHPs are associated with increased intracellular $Ca^{2+}$, mediated in part by $Ca^{2+}$ entry via L-type channels [14, 18, 20, 21]. Aged animals show increases in L-type channel expression and/or channel density at the plasma membrane [22–25]. Animals with higher $Ca^{2+}$ channel density perform poorly in spatial tasks [23], while blockers of L-type channels can restore learning and plasticity in older animals [26, 27].

It is not well understood how changes in ion channel gene expression and hippocampal PC excitability may affect neuron responsiveness and microcircuit output. In part, this is due to challenges inherent in performing the needed experiments. Single PCs are difficult to access in intact animals where hippocampal microcircuit function is preserved. It is also difficult to tease apart the influence of the many different neurophysiological factors that change during aging. Mathematical modeling provides a means to understand more about the effects of aging on hippocampal cellular excitability by controlling factors we cannot control experimentally.

Our previous work shows that mathematical expressions for different passive and active ion transport mechanisms can be derived from first principles of thermodynamics [28, 29] using a common functional form [30]. This results in a realistic representation of ionic flow across the membrane, and allows the model to reproduce phenomena such as rectification of ion currents seen in recordings. We present a model that reproduces the diversity of firing patterns in CA1 PC recordings, including adaptive firing, stimulus-induced bursting, and spontaneous bursting [31]. In addition, we reproduce several electrophysiological characteristics of aging by varying the expression of $Ca^{2+}$ channels in the model, and make predictions about bursting activity in aged CA1 PCs, which to our knowledge has not been reported. We believe this model is ideal to further study the effects of various biophysical changes in CA1 PCs during aging, as well as potentially forming the basis for biophysical, yet computationally inexpensive, network models.

## Materials and methods

### Model

To simulate the electrical activity of CA1 PCs, we used an extended version of a two-dimensional, biophysical model previously developed and characterized by two of the present authors [28–30]. The present model differs from our previous formulations in that it includes $Ca^{2+}$ dynamics, i.e. it is three-dimensional, allowing for additional behaviors like bursting [32]. In addition, it is specially tuned by incorporating specific channel variants and corresponding

experimental data from hippocampal PCs, as described below. The equations for the ionic currents are derived from first principles of thermodynamics. Previous modeling studies have shown that to reproduce firing behaviors such as spike frequency adaptation and bursting, the minimum number of variables is three [32, 33]. In particular, $Ca^{2+}$ dynamics are important for producing adaptation and burst firing in CA1 PCs (for review see [31]). The model dynamics are therefore described by three ordinary differential equations for the time-dependent changes in the transmembrane potential ($v$, in mV), the proportion of open $K^+$ channels ($w$ in [0, 1]), and the intracellular $Ca^{2+}$ concentration ($c$, in $\mu$M), respectively [30]. Based on a well-known relationship between voltage-dependent activation of delayed rectifier $K^+$ channels and inactivation of $Na^+$ channels, $w$ also represents the proportion of inactivated $Na^+$ channels [34, 35].

It is assumed that the membrane potential changes due to currents produced by ions transported across the membrane. We take into account currents mediated by a voltage-gated inactivating $Na^+$ channel ($I_{NaT}$), voltage-gated L-type $Ca^{2+}$ channel ($I_{CaL}$), voltage- and $Ca^{2+}$-gated $K^+$ channels ($I_{DK}$ and $I_{SK}$, respectively), and a $Na^+/K^+$-ATPase ($I_{NaK}$). We also incorporate a forcing term ($I_F$), which can be used to stimulate the model PC and is explained in more detail below. The time-dependent change in membrane potential can be written as

$$C_m \partial_t v = I_F - I_{NaT}(v, w) - I_{CaL}(v, c) - I_{DK}(v, w) - I_{SK}(v, c) - I_{NaK}(v). \tag{1}$$

with $\partial_t$ representing the instantaneous change with respect to time. $C_m$ (pF) is a constant representing the change in the density of charge around the membrane with respect to voltage, typically referred to as membrane capacitance in models based on electrical circuits [36]. Based on recordings from rat CA1 PCs, it is assumed that $C_m$ = 25 pF [37].

All the currents in Eq (1) are modeled using the same generic functional form, a product

$$I_x = a_x G_x \varphi_x, \tag{2}$$

$x \in \{NaT, CaL, DK, SK, NaK\}$, where $a_x$ is a whole-cell current amplitude (pA), $G_x$ is a gating term (between 0 and 1), and $\varphi_x$ is an adimensional term describing the driving force for the transmembrane flux (Table 1). The terms $a_x = s_x N_x$, $x \in \{NaT, CaL, DK, SK, NaK\}$ are whole-cell current amplitudes with $s_x$ (pA) representing the current flowing through a single channel (or pump), and $N_x$ representing the number of membrane proteins mediating the current (e.g. number of $K^+$ channels). Of interest, $s_x$ is $\sim$1 pA for most voltage-gated channels [38], and is $\sim$5–10 pA for SK channels [39].

Assuming that none of the currents in the model exhibit rectification [30], which agrees with recordings of the included currents, the adimensional component of the transmembrane

**Table 1. Transport mechanisms included in the model.**

| Transport mechanism | Current | Amplitude ($a$) | Gating ($G$) | Flux $\varphi$ |
|---|---|---|---|---|
| Transient $Na^+$ channels | $I_{NaT}(v, w)$ | $a_{Na}$ | $S_m(v)(1-w)$ | $\varphi_{Na}(v)$ |
| L-type $Ca^{2+}$ channels | $I_{CaL}(v, c)$ | $a_{Ca}$ | $S_n(v)$ | $\varphi_{Ca}(v)$ |
| Delayed rectifier $K^+$ channels | $I_{DK}(v, w)$ | $a_{DK}$ | $w$ | $\varphi_K(v)$ |
| SK $Ca^{2+}$-dependent $K^+$ channels | $I_{SK}(v, c)$ | $a_{SK}$ | $H_{SK}(c)$ | $\varphi_K(v)$ |
| $Na^+/K^+$ pumps | $I_{NaK}(v)$ | $a_{NaK}$ | 1 | $\varphi_{NaK}(v)$ |

All ion fluxes are given by a product of the form $I_x = a_x G_x \varphi_x$, where $a_x$, $G_x$, and $\varphi_x$ represent, respectively, the amplitude (normalized by membrane capacitance), gating, and driving force terms for the flux. $G_{NaK} = 1$ represents saturation of the $Na^+/K^+$ pumps. Note that inactivation of $Na^+$ channels is also represented by $w$ [34, 35], so the proportion of non-inactivated $Na^+$ channels is $1 - w$.

flux can be simplified and written as

$$\varphi_x(v) = 2\eta_x \sinh\left(\eta_x \frac{v - v_x}{2v_T}\right), \tag{3}$$

where $\eta_x$ represents the number of charges transported across the membrane in a single transport event, and $v_x$ is the reversal potential for the current, $x \in \{NaT, CaL, DK, SK, NaK\}$. For channels, $v_x$ is the Nernst potential for the ion [30]. Of note, $\eta_x = 1$ for $x \in \{DK, SK, NaK\}$ and $\eta_{NaT} = -1$ [30], which gives

$$\varphi_x(v) = 2 \sinh\left(\frac{v - v_x}{2v_T}\right), \tag{4}$$

for $x \in \{NaT, DK, SK, NaK\}$. For $Ca^{2+}$ channels, the total charge transported by one ion crossing the membrane from the extracellular space is $\eta_{CaL} = -2$, so

$$\varphi_{Ca}(v) = 4 \sinh\left(\frac{v - v_{Ca}}{v_T}\right). \tag{5}$$

The driving force for flux is assumed to be the same for DK and SK channels. Therefore, the label $K$ is used for both fluxes from here on. The thermal potential $v_T = kT/q$ (mV), where $k$ is Boltzmann's constant (mJ/°K), $T$ is the absolute temperature (°K), and $q$ is the elementary charge (Coulombs). The Boltzmann constant can be thought of as a scaling factor between macroscopic (thermodynamic temperature) and microscopic (thermal energy) physics [40]. The reversal potentials for the different currents depend on the Nernst potentials for each ion, as given by

$$v_x = \frac{v_T}{z_x} \ln\left(\frac{[x]_o}{[x]_i}\right), \quad x \in \{Na, Ca, K\} \tag{6}$$

where $z_x$ is the ion valence and $[x]_o$ and $[x]_i$ are the ion concentrations outside and inside the cell, respectively. The reversal potential for the $Na^+/K^+$-ATPase is given by $v_{NaK} = v_{ATP} + 3v_{Na} - 2v_K$ [30]. The Nernst potentials for $Na^+$ and $K^+$ are assumed to be constant, but $v_{Ca}$ varies because the intracellular $Ca^{2+}$ concentration is a state variable in the model.

**Gating.** The auxiliary functions describing voltage-dependent activation are given by

$$S_j(v) = \frac{\exp\left(g_j \frac{v - v_j}{v_T}\right)}{1 + \exp\left(g_j \frac{v - v_j}{v_T}\right)}, \quad j \in \{m, n, w\}, \tag{7}$$

where $g_j$ controls the steepness of the activation curve for $Na^+$ ($m$), $Ca^{2+}$ ($n$), or DK ($w$) channels, and $v_j$ represents the half-activation voltage for those channels. The function

$$R_w(v) = r_w\left[\exp\left(b_w g_w \frac{v - v_w}{v_T}\right) + \exp\left((b_w - 1)g_w \frac{v - v_w}{v_T}\right)\right], \tag{8}$$

describes the voltage-dependence of the rate of activation of the DK channels. The parameters $r_w$ and $b_w$ represent the recovery rate and the asymmetry in the gating relative to voltage that biases the time constant for the gating process, respectively.

The dynamics for the proportion of activated DK channels, $w$, are assumed to be logistic,

$$\partial_t w = w(S_w(v) - w)R_w(v), \tag{9}$$

which yields better fits and is more consistent with the dynamics of activation in channel populations recorded in voltage-clamp experiments (e.g., see the activation curves in [41–43]).

The gating of the SK channel is not voltage-dependent, but instead depends on intracellular $Ca^{2+}$ binding. Its activation is modeled using a Hill equation that depends on the intracellular concentration of $Ca^{2+}$, as used to fit data from channel recordings [44]:

$$H_{SK}(c) = \frac{c^2}{c^2 + c_{SK}^2},$$ (10)

where $c_{SK}$ represents the half-activation $Ca^{2+}$ concentration for the SK channels, with a reported value of 0.74 $\mu$M (740 nM) [39, 44].

For the dynamics of intracellular $Ca^{2+}$, we assume recovery toward a steady state $c_\infty$ at a rate $r_c$, with increments caused by the $Ca^{2+}$ current $I_{Ca}$ [35],

$$\partial_t c = r_c(c_\infty - c) - \tilde{k}_c I_{CaL}(v, c).$$ (11)

The term $\tilde{k}_c$ in Eq (11) is a conversion factor ($\mu$M/pCoul) that accounts for the effect of $Ca^{2+}$ flux across the membrane on the intracellular $Ca^{2+}$ concentration.

The term $I_F$ represents a stimulus *forcing* the membrane; that is, current from an electrode ($I_{Stim}$), or time-dependent fluctuations from the local field potential (LFP). LFP activity is simulated by replacing the term $I_F$ with a time-dependent, Ornstein-Uhlenbeck (OU) process with amplitude $a_F(t)$ (pA). The mean is represented by $\mu_F$ (pA) (drift term) [45] given by [46]

$$a_F(t + \delta) \;\; = \;\; a_F(t)\left(1 - \frac{\delta}{\tau_F}\right) + \left[\mu_F\delta + \eta(t)\;\sqrt{d_{Stim}\delta}\right],$$ (12)

where $\delta$ is a small time step, $\tau_F$ is a relaxation time, and $\eta(t)$ is an independent white noise process with zero-mean and unit standard deviation. In our simulations, the mean is set close to the rheobase for the model PCs ($\sim 50$ pA). The process has a variance $\sigma_F^2 = d_F\delta/2$ (pA), which means $d_F$ can be approximated if an estimation of the variance of the current $a_F$ is available [47, 48].

**Change of variables to obtain numerical solutions.**   To simplify the numerics, we change variables

$$u = v/v_T,$$ (13)

and adjust all voltages accordingly as

$$u_l = v_l/v_T, \quad l \in \{NaT, CaL, K, NaK, m, n, w\}.$$ (14)

The new equation for the normalized voltage is

$$\partial_t u = \frac{\partial_t v}{v_T}.$$ (15)

To simplify the notation and reduce the number of operations during the numerical integration, we also reparametrize the amplitudes as

$$A_l = \frac{2a_l|\eta_l|}{v_T\,C_m},$$ (16)

for $l \in \{NaT, CaL, K, NaK\}$, in units of 1/ms. Similarly, the activation functions for the Na$^+$,

$Ca^{2+}$, and DK currents can also be rewritten respectively as

$$f_\infty(u) = \{1 + \exp[g_f(u_f - u)]\}^{-1}, \quad f \in \{m, n, w\}. \tag{17}$$

The result is a new equation of the form

$$\partial_t u = J_F - A_{NaT}(1 - w)m_\infty(u)\varphi_{Na}(u) - A_{CaL}n_\infty(u)\varphi_{Ca}(u, c) - \\ (A_{DK}w + A_{SK}H_{SK}(c))\varphi_K(u) - A_{NaK}\varphi_{NaK}(u). \tag{18}$$

The term $J_F$ (1/ms) is the input current $I_F$ (pA) divided by $v_T C_m$. After the change in variables and the normalization of the current amplitudes, Eq (11) changes to

$$\partial_t c = r_c(c_\infty - c) - k_c A_{CaL}n_\infty(u)\varphi_{Ca}(u, c), \tag{19}$$

where $k_c = \tilde{k}_c v_T C_m$.

## Parameters

The currents were modeled to fit as closely as possible the biophysical properties of those carried by channel variants expressed in mammalian neurons, and specifically CA1 PCs, where data are available. The DK current is based on that mediated by $K_v 2.1$ channels, the predominant channel underlying the delayed rectifier current in rat hippocampal neurons [49]. The L-type $Ca^{2+}$ current is based on that carried by $Ca_v 1.2$ (class C) channels, the predominant L-type channel isoform expressed in rat brain [50]. Additional details about the parameters can be found in Table 2.

Wherever possible, model parameters were taken from studies in rodent (mice and rat) hippocampal CA1 PCs. If data were not available, we obtained parameters from other types of mammalian cell, or from studies of mammalian ion channels in expression systems like *Xenopus* oocyte. Physical constants and other parameters we would not expect to vary, such as the intra- and extracellular concentrations of ions or the cellular capacitance, were fixed. Biophysical properties of the ion channels, such as their half-activation voltages, were also fixed. The parameters we varied were primarily those corresponding to maximum current amplitudes, which can change acutely due to modulation or channel phosphorylation [51, 52], or chronically due to changes in ion channel expression that occur with age [22, 24].

By exploring the model through parameter variations, we were able to find parameter sets that produced different firing patterns, such as adaptive firing, conditional bursting, and spontaneous bursting. The rationale behind finding parameters for qualitatively different firing patterns that emerge in a three-dimensional model similar to this one was established by Av-Ron and colleagues [32]. For example, they show that one way to observe transitions between adaptive firing and bursting is to vary the ratio of DK to $Ca^{2+}$-dependent $K^+$ channels. Therefore, in our study, we began by setting the model parameters within the base physiological range given by experimental recordings in PCs, and then took into consideration the different ratios of select parameters and their effects. Once we defined these different restricted parameter ranges, we then tuned semi-manually, i.e. using Python for loops to quickly run through a series of parameter values using knowledge of how increases or decreases in specific parameters should change the firing of cells (e.g., a larger SK current leads to greater inhibition of firing, lower firing frequencies, etc.). The specific PC firing patterns are described in more detail in the Results section, but the respective parameter sets are included in Table 3 for ease of comparison.

**Table 2. Constants and parameters.**

| parameter | description | value | units | reference |
|---|---|---|---|---|
| $k$ | Boltzmann's constant | $1.381e^{-20}$ | mJ/K | physical constant [38] |
| $q$ | elementary charge | $1.602e^{-19}$ | C | physical constant [38] |
| $T$ | absolute temperature | $273.15 + 37$ | K | adjusted to mammalian body temperature of 37˚C [38] |
| $a_{NaT}$ | amplitude of transient Na$^+$ current | 1000–2300 | pA | set to produce currents of $\sim$2–7 nA, in range recorded in CA1 PCs from rats [53] and guinea pigs [54] |
| $a_{CaL}$ | amplitude of L-type Ca$^{2+}$ current | 25 or 50 | pA | set to produce currents of $\sim$2–3 nA or $\sim$5–6 nA as recorded in young and aged CA1 PCs, respectively [20] |
| $a_{DK}$ | amplitude of delayed rectifier K$^+$ current | 6000–8000 | pA | set to produce currents of $\sim$6–9 nA, in range recorded from HEK cells expressing rat Kv2.1 and $I_K$ in hippocampal neurons [55] |
| $a_{SK}$ | amplitude of Ca$^{2+}$-dependent K$^+$ current | 300–1600 | pA | set to produce currents of $\sim$100–800 pA, depending on Ca$^{2+}$ concentration, as recorded in SK-transfected cells [56] |
| $a_{NaK}$ | amplitude of Na$^+$/K$^+$-ATPase current | 10–23 | pA | set to produce currents of $\sim$70–190 pA, similar to but on high end of range recorded in hippocampal PCs [57] |
| $\eta_x$ | charge moved across the membrane in a single transport event $x \in \{NaK, DK, SK\}$ | 1 | – | 1 net positive charge moving outward [30] |
| $\eta_{NaT}$ | charge transported across the membrane by $NaT$ channels | -1 | – | 1 net positive charge moving inward [30] |
| $\eta_{CaL}$ | charge transported across the membrane by Ca channels | -2 | – | 2 net positive charges moving inward [30] |
| $v_{Na}$ | Nernst potential for Na$^+$ | 60 | mV | in range reported for mammalian cells [58] |
| $[Ca]_o$ | extracellular Ca$^{2+}$ concentration (intracellular varies) | 1.5 | mM | in range reported for mammalian neural tissue [59] |
| $v_{Ca}$ | Nernst potential for Ca$^{2+}$ | variable; baseline $\sim$128 | mV | in range reported for mammalian cells [58]; varies since intracellular Ca$^{2+}$ concentration is a model variable |
| $v_K$ | Nernst potential for K$^+$ | -89 | mV | in range reported for mammalian cells [58] |
| $v_{ATP}$ | Nernst potential for ATP | -420 | mV | value used in model of mammalian heart cells and based on fit to data [60] |
| $v_{NaK}$ | Nernst potential for Na$^+$/K$^+$-ATPase | -62 | mV | calculated based on the Nernst potentials for ATP, Na$^+$, and K$^+$, and a 3:2 stoichiometry, respectively [61]; $v_{NaK} = 3v_{Na} - 2v_K - v_{ATP}$ |
| $r_w$ | rate of activation of delayed rectifier K$^+$ current | 1.0–1.8 | ms | fit so that the duration of the action potential is approximately 2 ms [62] |
| $s_w$ | asymmetry of time constant of delayed rectifier K$^+$ current | 0.3 | - | based on fit; if higher (0.5–0.7) APs are the wrong shape and do not ride on sufficient plateau potential compared to recordings |
| $v_m$ | half-activation potential of Na$^+$ current | -19 | mV | in range reported for transient Na$^+$ channels in CA1 PCs [63, 64] |
| $v_n$ | half-activation potential of Ca$^{2+}$ current | 3 | mV | in range recorded for high-voltage activated Ca$^{2+}$ currents in rat CA1 PCs [65]; see also recordings from oocytes [66] or HEK cells [67] expressing Ca$_v$1.2 channels |
| $v_w$ | half-activation potential of delayed rectifier K$^+$ current | -1 | mV | in range reported for rat Kv2.1 channels expressed in COS-1 cells [49] |
| $c_{SK}$ | half-activation Ca$^{2+}$ concentration for SK current | $7.4e^{-4}$ | mM | based on recordings from oocytes expressing rat SK channel variant [44] |
| $g_m$ | activation slope of Na$^+$ current | 5.0 | - | in range reported for Na$^+$ current in mouse [68] and rat [69] CA1 neurons |
| $g_n$ | activation slope of Ca$^{2+}$ current | 5.0 | - | in range reported for Ca$_v$1.2 expressed in HEK cells [70] |
| $g_w$ | activation slope of delayed rectifier K$^+$ current | 3.8 | - | fit to data from rat brain delayed rectifier channels [71] |
| $c_\infty$ | minimum intracellular Ca$^{2+}$ concentration | $1e^{-4}$ | mM | equivalent to 100 nM, approximate resting intracellular Ca$^{2+}$ concentration in rat CA1 PCs [17, 72, 73] |
| $r_c$ | intracellular Ca$^{2+}$ removal rate constant | $1e^{-3}$ to $5e^{-3}$ | ms$^{-1}$ | adjusted to produce Ca$^{2+}$ dynamics as recorded in rat CA1 PCs [72] |
| $k_c$ | conversion factor to calculate effect of Ca$^{2+}$ current on intracellular Ca$^{2+}$ concentration | $3e^{-6}$ to $6e^{-6}$ | mM | adjusted to produce Ca$^{2+}$ dynamics as recorded in rat CA1 PCs [72] |

**Table 3. Parameters used to produce different firing patterns in young PC.**

| parameter | adaptive firing | conditional bursting | spontaneous bursting |
|---|---|---|---|
| $a_{NaT}$ ($A_{NaT}$) | 1000 (2.99) | 1300 (3.89) | 2300 (6.88) |
| $a_{CaL}$ ($A_{CaL}$) | 25 (0.15) | 25 (0.15) | 25 (0.15) |
| $a_{DK}$ ($A_{KD}$) | 8000 (23.95) | 6000 (17.96) | 7000 (20.95) |
| $a_{SK}$ ($A_{SK}$) | 1400 (4.19) | 1600 (4.79) | 300 (0.90) |
| $a_{NaK}$ ($A_{NaK}$) | 10.0 (0.03) | 13 (0.04) | 23 (0.07) |
| $r_w$ | 1.0 | 1.8 | 1.1 |
| $r_c$ | $1e^{-3}$ | $5e^{-3}$ | $5e^{-3}$ |
| $k_c$ | $3e^{-6}$ | $6e^{-6}$ | $6e^{-6}$ |

Original amplitudes ($a_x$) are in pA. Reparametrized amplitudes ($A_x$) in parentheses are calculated by $2 * \frac{a_x}{v_T C_m}$ for $x \in \{NaT, DK, SK, NaK\}$ and $4 * \frac{a_{CaL}}{v_T C_m}$ for $Ca^{2+}$ channels. For all calculations, $v_T C_m$ = 668.171 mV pF.

## Simulations

All code was written in Python 3.7.4 and run on MacBook Pro laptops with 2.9 GHz Intel Core i5 processors. Simulations were performed using functions from the Python library NumPy [74]. Figures were produced with the Python library Matplotlib [75]. OU processes were simulated using the pyprocess package [76].

## Resource availability

Resources generated by this study (code, figures, and manuscript files) are available via GitHub (github.com/emckiernan/agingCA1) and archived via Zenodo (doi.org/10.5281/zenodo.6788229). To facilitate reuse, resources are shared under open licenses (see the license file in our GitHub repository). To promote reproducibility, Python code is embedded in a Jupyter notebook [77] that explains the code, how to use it, and how to generate all figures, including those in the Supporting Information.

## Study design

While aged cells display a number of biophysical changes, we focused on their $Ca^{2+}$ channel expression. Aged CA1 PCs show an increase in the number of functional transmembrane L-type $Ca^{2+}$ channels [22–24]. In particular, CA1 PCs from aged rats have increased expression of $Ca_v1.2$ at the plasma membrane [25]. With these results in mind, we decided to simulate one aspect of aging by changing the number of $Ca_v1.2$ channels in our model membrane. We asked the question, is a change in $Ca_v1.2$ expression sufficient to reproduce the various changes in excitability, such as increased spike frequency adaptation, observed experimentally in aged CA1 PCs? In addition, CA1 PCs are known to burst [31], but we are not aware of studies comparing their bursting patterns in young versus aged animals. Therefore, we used our model to also investigate the effects of altered $Ca_v1.2$ channel expression on bursting activity. In all the following simulations, once the models have been tuned to a specific firing pattern, young and aged model PCs (hereafter referred to as yPC and aPC, respectively) are identical with respect to every parameter except the maximum amplitude of their L-type $Ca^{2+}$ current, which is set to produce currents of $\sim$2–3 nA or $\sim$5–6 nA to match the magnitude of currents seen in recordings of CA1 PCs from young and aged animals, respectively [20].

## Results

CA1 PCs display diverse firing patterns, ranging from repetitive adaptive spiking to stimulus-induced or spontaneous bursting (for review see [31]). Thus, to represent these cells, our model must reproduce this diversity, as well as age-related effects on firing reported in the literature.

### Modeling age-related changes in spike frequency adaptation

Many CA1 PCs respond to square-pulse current injection by firing several early spikes followed by adaptation which slows the frequency of firing [17, 18, 78]. To generate this firing pattern, we set the ionic currents to be the same amplitude range as observed in recordings of young adult CA1 PCs, with amplitudes for the $Na^+$ and $Ca^{2+}$ currents at $\sim$2–3 nA, the DK current approximately double the inward cationic currents, and the SK current at $\sim$400–700 pA, depending on the intracellular $Ca^{2+}$ concentration (see Tables 2 and 3). This balance of ionic currents successfully generates adaptive firing similar to recordings (Fig 1A and S2 Fig).

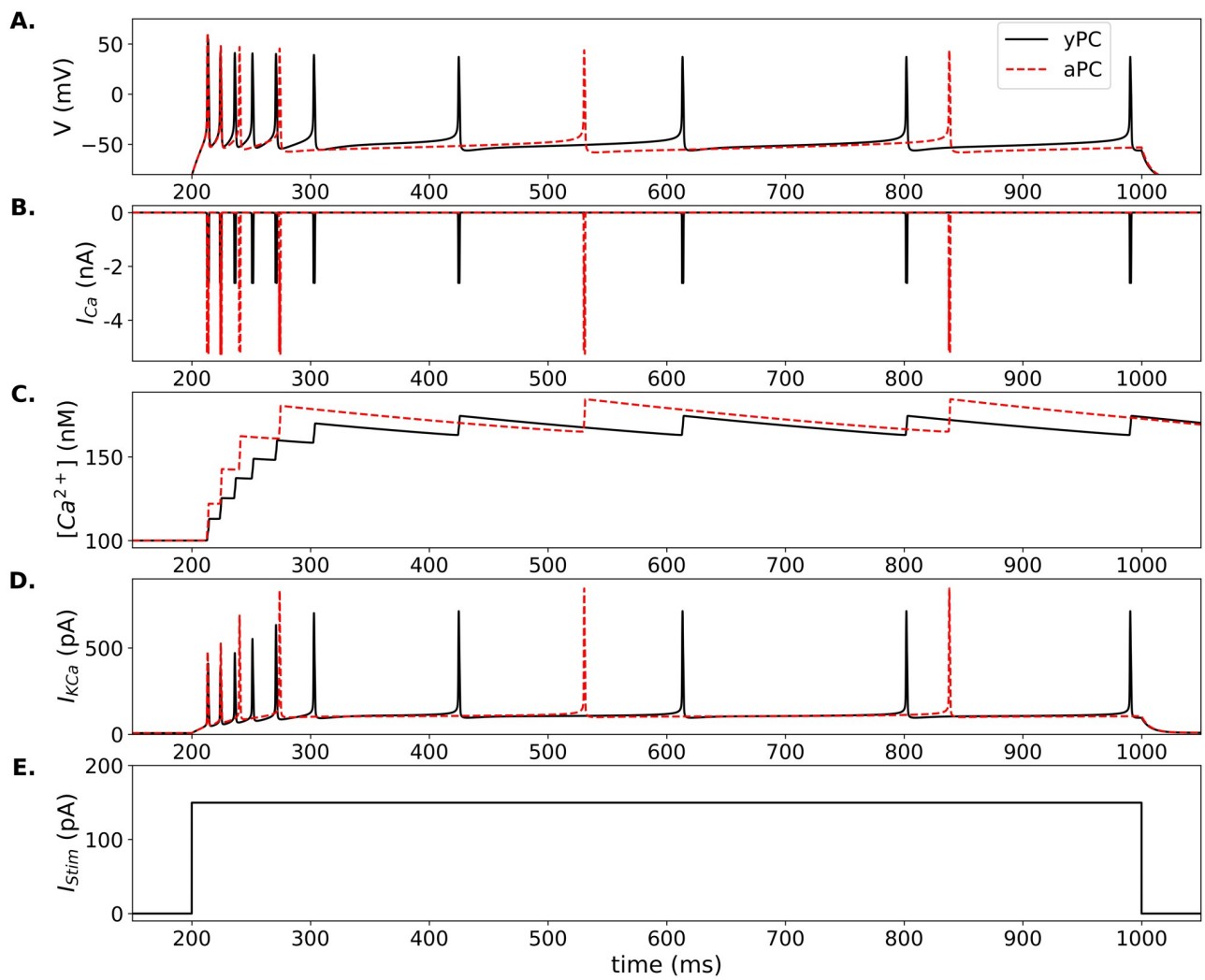

**Fig 1. Adaptive firing in model PCs.** (A.) Adaptive firing in the yPC (solid black traces) versus aPC (dashed red traces) in response to a 800 ms 150 pA square-pulse stimulation shown in (E.). Corresponding $Ca^{2+}$ currents, intracellular $Ca^{2+}$ concentration, and SK currents are shown in (B.), (C.), and (D.), respectively. Parameters for yPC: $a_{NaT} = 1000$, $a_{CaL} = 25$, $a_{DK} = 8000$, $a_{SK} = 1400$, $r_w = 1.0$, $r_c = 1e^{-3}$, and $k_c = 3e^{-6}$. All parameters for aPC the same except $a_{CaL} = 50$. Parameter units, and additional parameters kept constant for all simulations, are in Table 2.

Studies show that adaptation is more pronounced in aged than young animal cells, leading to a shorter initial period of fast spiking, followed by fewer spikes or complete cessation of firing [17–19, 79]. To compare the young (yPC) and aged (aPC) model cells, all parameters were fixed except for the maximum amplitude of the L-type $Ca^{2+}$ current, which was set to produce currents of $\sim$2–3 nA (young) or $\sim$5–6 nA (aged), based on recordings [20]. This difference in the $Ca^{2+}$ current causes the rate of firing in the first $\sim$100 ms (early firing) to decrease from 60 Hz in the yPC to 40 Hz in the aPC (Fig 1A and S4A Fig). Firing for the remaining stimulation time (late firing) is also affected, decreasing from $\sim$6 Hz to $\sim$3 Hz in the yPC versus aPC, respectively (Fig 1A and S4B Fig). The effect and frequencies are similar to those seen in recordings of CA1 PCs in young and old rabbits [18].

Examining the $Ca^{2+}$ and SK dynamics during the response reveals the mechanisms underlying the stronger adaptation in the aPC (Fig 1B–1D). The first two spikes occur nearly simultaneously in the two model cells. However, the larger increase in intracellular $Ca^{2+}$ in the aPC induces a larger SK current, which in turn slows the cell's firing. The aPC falls behind the yPC by the third spike, and then slows its firing further as the response continues.

## Modeling age-related changes in AHPs

AHP generation has been studied in CA1 PCs [80], particularly in the context of aging [17, 18, 79, 81]. To induce AHPs, we kept the same parameters as in the previous simulations. We then stimulated model PCs with a 100 ms square pulse of sufficient amplitude to generate a burst of 4 APs (Fig 2). The AHPs produced under these conditions in the yPC have a peak amplitude of 3–4 mV (S5 Fig inset), similar to recordings [14, 81]. The aPC required 35 pA more current than the yPC to fire the same number of spikes (Fig 2E). However, the aPC fires earlier than the yPC due to its increased $Ca^{2+}$ current (Fig 2A inset). The aPC generates an AHP 1–2 mV larger than seen in the yPC (Fig 2B), similar to the difference observed in recordings between young and aged cells [14, 17, 18]. In the model, this larger AHP is due to an increased accumulation of $Ca^{2+}$ in the aPC, which in turn produces a larger SK current (Fig 2C and 2D, respectively).

## Modeling age-related changes in burst firing

**Bursting in response to stimulation.** Some CA1 PCs fire bursts instead of trains of spikes [82], especially in certain developmental periods [83]. Burst firing can be generated in the model with several different parameter combinations. For the following simulations, we modified several of the current amplitudes and select kinetics, all within physiological limits (see Table 3). Under this parameter regime, model PCs are silent at rest but burst if stimulated (Fig 3 and S6 Fig).

To explore the effects of aging on bursting, we fixed all parameters except for the maximum $Ca^{2+}$ current amplitude, as previously. We then stimulated the two model PCs with a series of square-pulse current injections of increasing amplitudes (50–170 pA) to compare their responses. Again, the larger $Ca^{2+}$ current in the aPC causes it to fire either sooner or nearly simultaneously with the yPC shortly after stimulus onset in all simulations (S8 Fig). However, the relative timing of the PCs' firing after the first burst depends on the stimulus amplitude (Fig 3).

At lower stimulation amplitudes (50 and 80 pA; Fig 3A and 3B), the aPC continues to burst sooner or nearly simultaneously, but always fires fewer spikes per burst than the yPC (2 versus 3, respectively). As the stimulation amplitude increases (110 pA; Fig 3C), the two PCs again fire nearly simultaneously at the onset. However, because the aPC fires fewer spikes per burst, it is able to recover sooner and burst before the yPC for four cycles. It is only towards the end of the stimulus that the larger AHP in the aPC eventually brings it into sync again with the yPC. Finally,

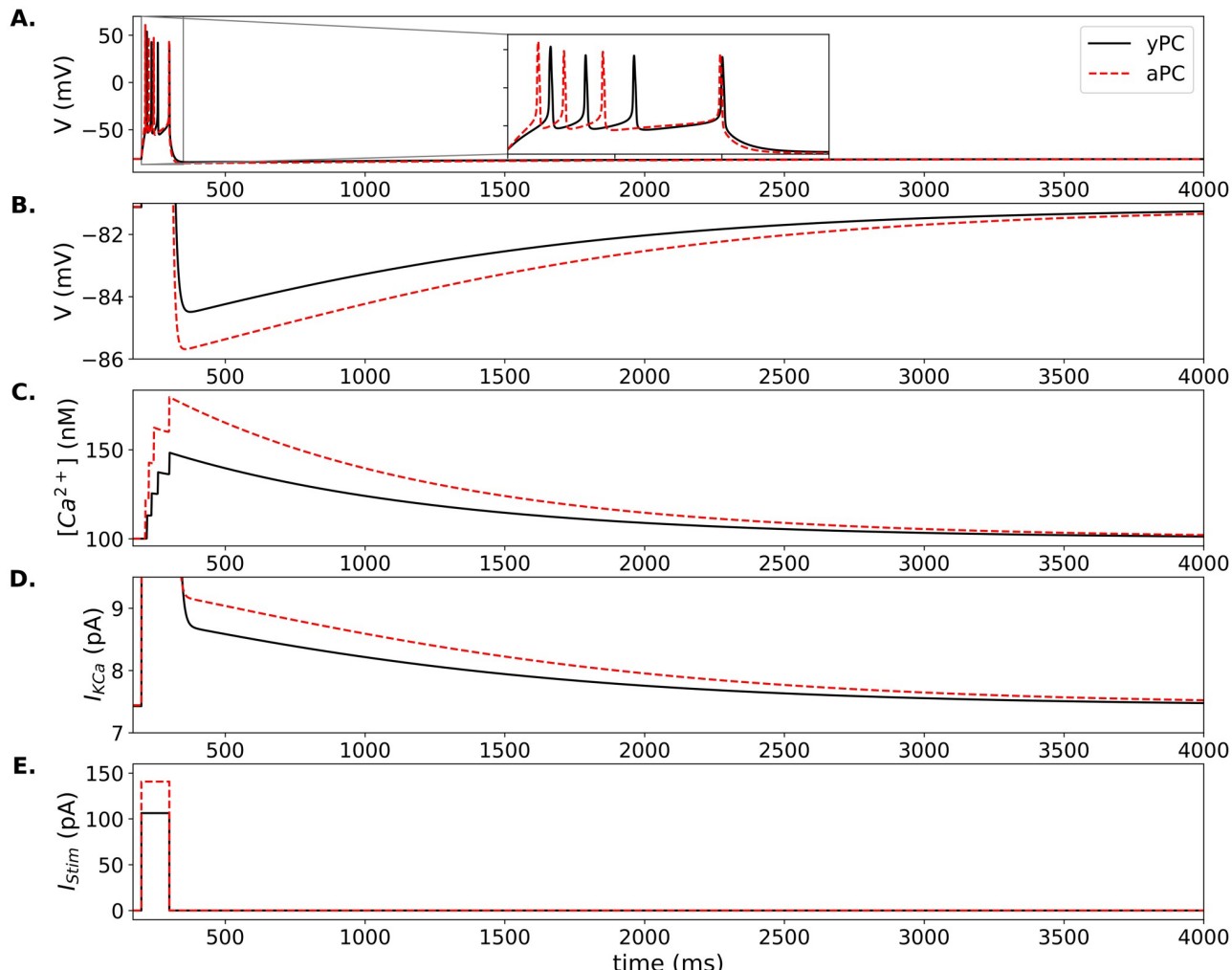

**Fig 2. AHPs in model PCs.** (A.) Responses of the yPC (solid black traces) and aPC (dashed red traces) to 100 ms pulse. Voltage zoom in (B.) shows AHPs in detail. Corresponding $Ca^{2+}$ concentrations and SK currents are shown in (C.), and (D.), respectively. Parameters same as in Fig 1, except for the current stimulation duration and amplitude. Current amplitude adjusted to the minimum required (within 1 pA) to produce 4 spikes in each cell, i.e. 106 pA in yPC and 141 pA in aPC, as shown in (E.).

at the highest stimulation amplitudes (140 and 170 pA; Fig 3D and 3E), both PCs show stronger adaptation, but the yPC maintains something akin to bursting, while the aPC 'devolves' to a pattern more like tonic spiking. Depending on what is considered the relevant electrical event—the single spike or the burst—the aPC shows an increased number of events relative to the yPC (13 spikes versus 8 burst-like events, respectively), but the normal bursting pattern is lost.

**Spontaneous bursting.** A small percentage of CA1 PCs fire bursts in the absence of stimulation [84]. To generate this pattern, we again modfied several of the current amplitudes and select kinetics, all within physiological limits (see Table 3). Under this parameter regime, the yPC fires spontaneous bursts at a frequency of ∼ 1 Hz with 3 spikes per burst (Fig 4C and S9 Fig), similar to recordings [85].

Increasing the $Ca^{2+}$ current, as previously, changed the spontaneous firing pattern (Fig 4). Additional interesting effects can be seen if we vary the DK current amplitude within the range previously used for simulations, 6000–8000 pA (6–8 nA). At the highest DK amplitude,

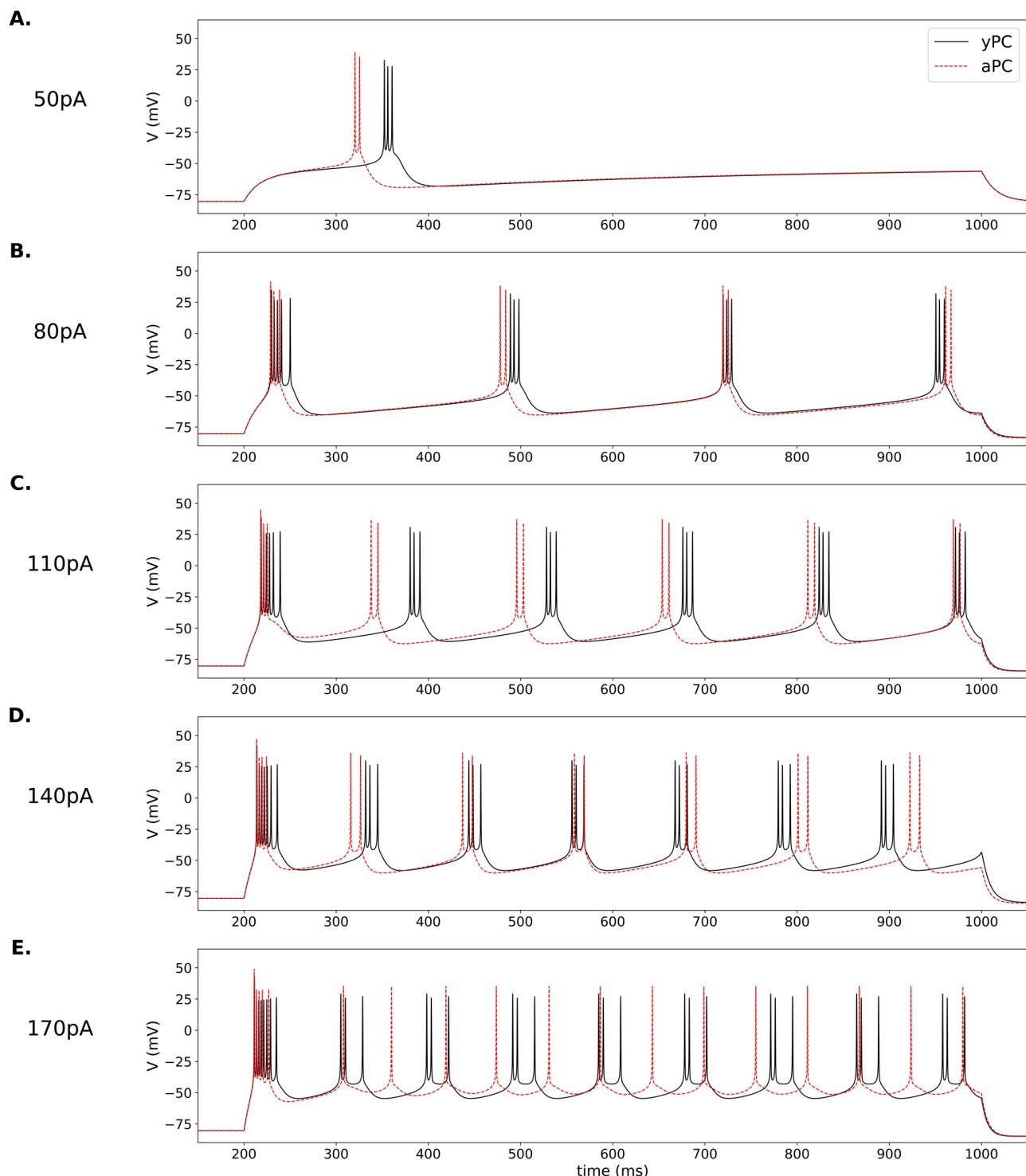

**Fig 3. Stimulated bursting in model PCs.** Bursting in the yPC (solid black traces) and aPC (dashed red traces) models in response to 800 ms current injections of 50–170 pA, as indicated in panels (A.)-(E.). Parameters for the yPC: $a_{NaT}$ = 1300, $a_{CaL}$ = 25, $a_{DK}$ = 6000, $a_{SK}$ = 1600, $r_w$ = 1.8, $r_c$ = $5e^{-3}$, $k_c$ = $6e^{-6}$. All parameters for the aPC the same except $a_{CaL}$ = 50.

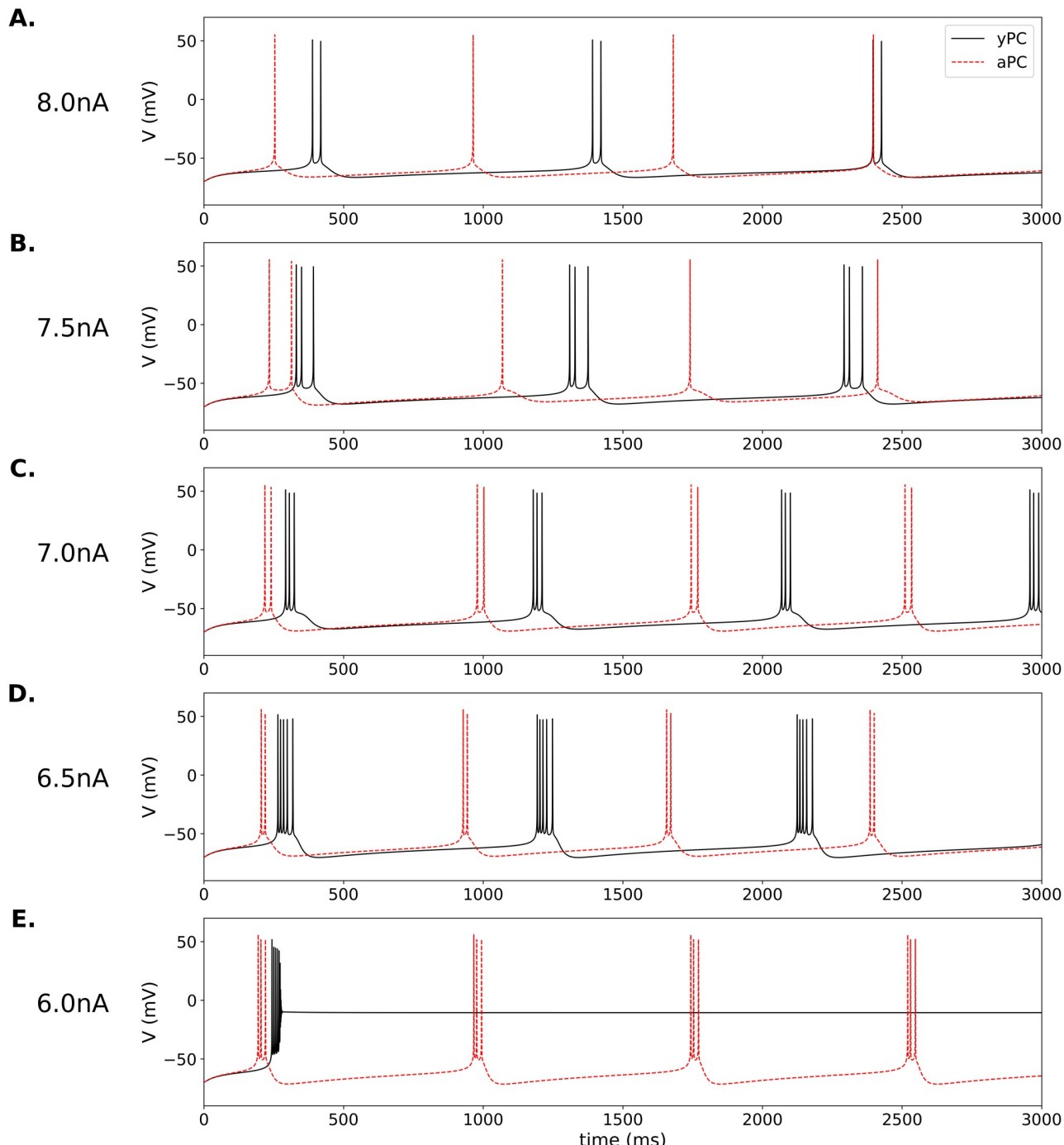

**Fig 4. Spontaneous bursting in model PCs.** Comparison of spontaneous electrical activity in the yPC (solid black traces) and aPC (dashed red traces) for different levels of DK current amplitude. $a_{DK}$ decreasing from 8000 to 6000 pA (8 to 6 nA) in steps of 500 pA, as indicated in (A.)-(E.). Parameters: $a_{NaT} = 2300$, $a_{CaL} = 25$, $a_{DK} = 7000$, $a_{SK} = 300$, $r_w = 1.1$, $r_c = 5e^{-2}$, $k_c = 6e^{-6}$, $I_F = 0.0$. All parameters for aPC the same except $a_{CaL} = 50$.

the yPC bursts spontaneously but with only 2 spikes per burst (Fig 4A). The aPC, however, does not burst but spikes tonically at a frequency of $\sim 2$ Hz. When the DK amplitude is reduced (7.5 nA), the yPC continues to burst, now with 3 spikes per burst, while the aPC still spikes tonically (Fig 4B).

Reducing the DK amplitude further (7 nA or 6.5 nA; Fig 4C or 4D), causes both PCs to burst spontaneously, though the aPC always fires fewer spikes per burst. One can use our Jupyter notebook to explore further, and see that while the $Ca^{2+}$ current is larger for the aPC, the maximum accumulation of intracellular $Ca^{2+}$ is either similar or higher in the yPC due to its additional spiking (S11 Fig). This causes a similar or larger SK current in the yPC, and results in a slower burst frequency.

Finally, reducing the DK amplitude to the lowest level (6.0 nA; Fig 4E) removes more of the 'brake' on the yPC and causes it to spike at high frequency and then quickly block depolarize. The aPC, on the other hand, retains the spontaneous bursting pattern, now with more spikes per burst.

## Responses to local field potential forcing

Square-pulse stimulation is useful for examining the timing of PC responses, but is not a physiologically realistic stimulus. Instead, to simulate local field potential (LFP) forcing onto CA1 PCs, we use an Ornstein-Uhlenbeck (OU) stochastic process [47, 48], as in Eq 12. First, we reset the model with the parameters needed to produce adaptive firing. In the yPC, LFP forcing produces repetitive, irregular firing at a frequency of $\sim 3$ Hz on average (Fig 5A), which is similar to recordings of spontaneous firing in CA1 PCs [86], particularly in response to certain types of activity in the surrounding electrical field [87, 88].

In response to the exact same LFP forcing applied to the yPC, the aPC with increased $Ca^{2+}$ current shows a similar irregular firing pattern, but slower frequency of $\sim 2$ Hz (Fig 5B; compare overlap in C). The simulation also shows several time points when the two cells fire almost simultaneously, and then the yPC fires again while the aPC does not. This apparent 'spike failure' has been seen in recordings of PCs from aged animals [13].

Next, we set the parameters to produce conditional bursting, as previously. The exact bursting pattern will vary depending on the stochastic OU process. However, under these conditions, LFP forcing in the yPC typically produces irregular burst firing at a frequency of $\sim 5$Hz (i.e. theta frequency), with 2–4 spikes per burst (Fig 6A). This firing pattern is similar to spontaneous activity recorded in a subset of CA1 PCs known as phasic theta-ON cells, which preferentially burst during theta activity recorded from the surrounding field [87, 89]. Increased $Ca^{2+}$ current in the aPC changes the firing pattern (Fig 6B). In response to LFP forcing, the aPC still fires irregular bursts, but with fewer spikes per burst (usually 3 max) and a higher occurrence of 2-spike bursts than seen in the yPC. In addition, the aPC fires single APs amidst the bursts, which occurs less frequently in the yPC under this parameter regime. Also, the timing of the bursts in the aPC can change relative to the yPC (see overlap in Fig 6C).

## Discussion

### Cellular heterogeneity

Our three-dimensional, single-compartment model derived from first principles of thermodynamics can reproduce the diversity of firing patterns recorded in CA1 PCs. Moving between the different patterns was achieved primarily by changes to the relative expression of ion channels in the model. We did not systematically explore the full parameter space, but future work could include bifurcation analysis to determine boundaries for each firing pattern. The flexibility of the model could be useful for researchers to study the effects of PC heterogeneity on network function. Geiller and colleagues [90] write, "Until very recently, hippocampus models and theories were built on a view of homogenous population of principal cells" (pg. 6). However, there is heterogeneity in CA1 PCs, especially during different development stages (for

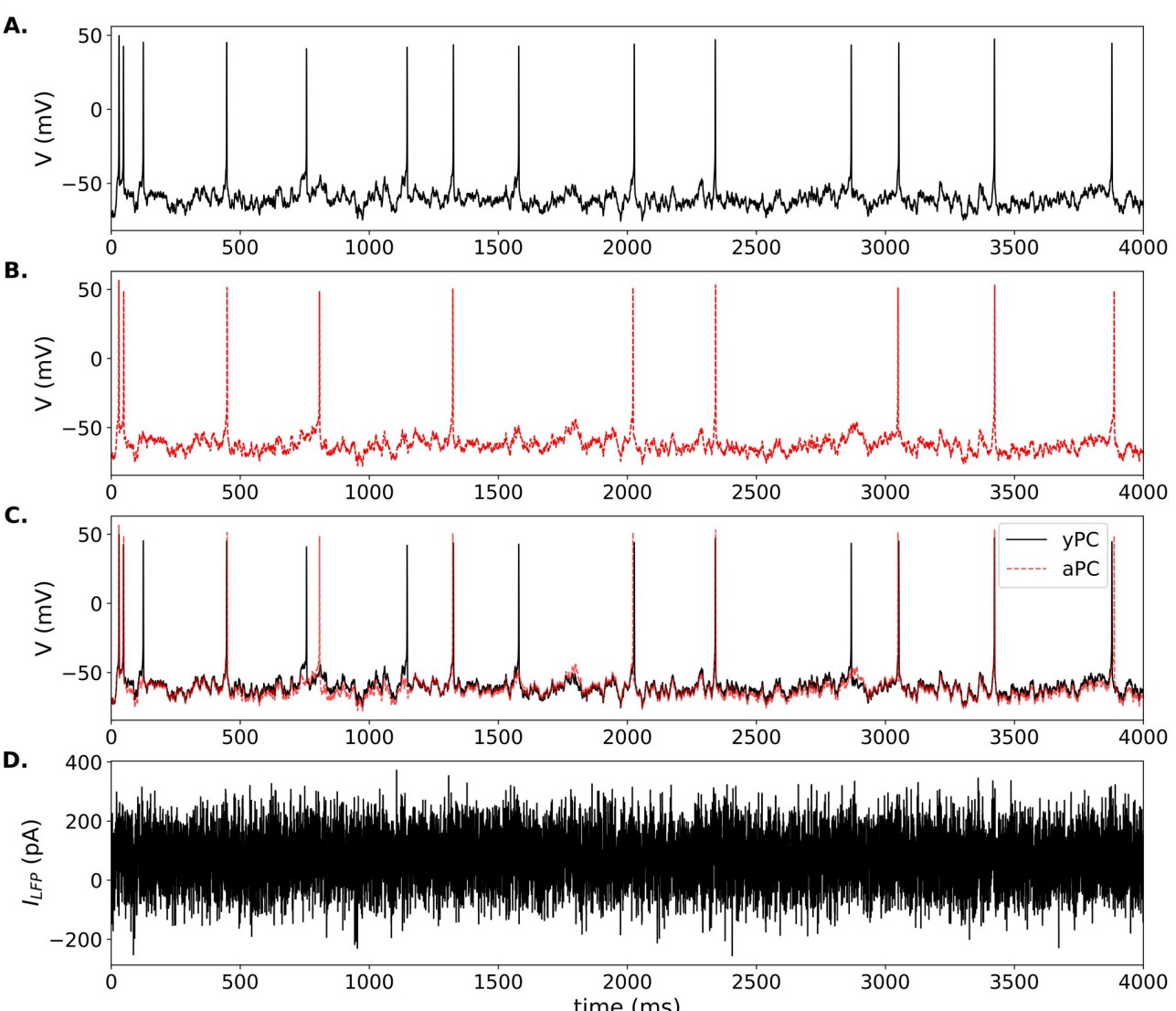

**Fig 5. LFP-induced firing in model PCs.** Responses of the yPC (solid black traces) and aPC (dashed red traces) to the same LFP forcing while in adaptive firing mode. (A.) yPC response, (B.) aPC response, and (C.) overlap of the two traces. Parameters for the yPC and aPC are the same as in Fig 1. LFP parameters: $\mu_F$ = 50.0 pA, $\sigma_F$ = 25.0 pA, $\tau_F$ = 1/2.0 for both model PCs.

review, see [31]). Lee and colleagues [91] write, "how the heterogeneous PCs integrate into the CA1 circuit remains unknown" (pg. 1129).

Experimentally, it is difficult to quantify how many PCs in a given network are displaying a specific firing pattern, and even harder, if not impossible, to manipulate these percentages. Furthermore, cells can transition between firing patterns [92], meaning percentages might fluctuate. With our minimal model, however, we could build small networks with different balances of adapting versus bursting PCs, and explore how this affects network output. This might help researchers understand the extent of 'acceptable' heterogeneity within hippocampal circuits, or how sensitive these networks are to changes in the overall balance of cells displaying different firing patterns. Such insights will be relevant to neurophysiological aging, as these balances can change over a lifespan [83], and also to age-related brain disorders such as

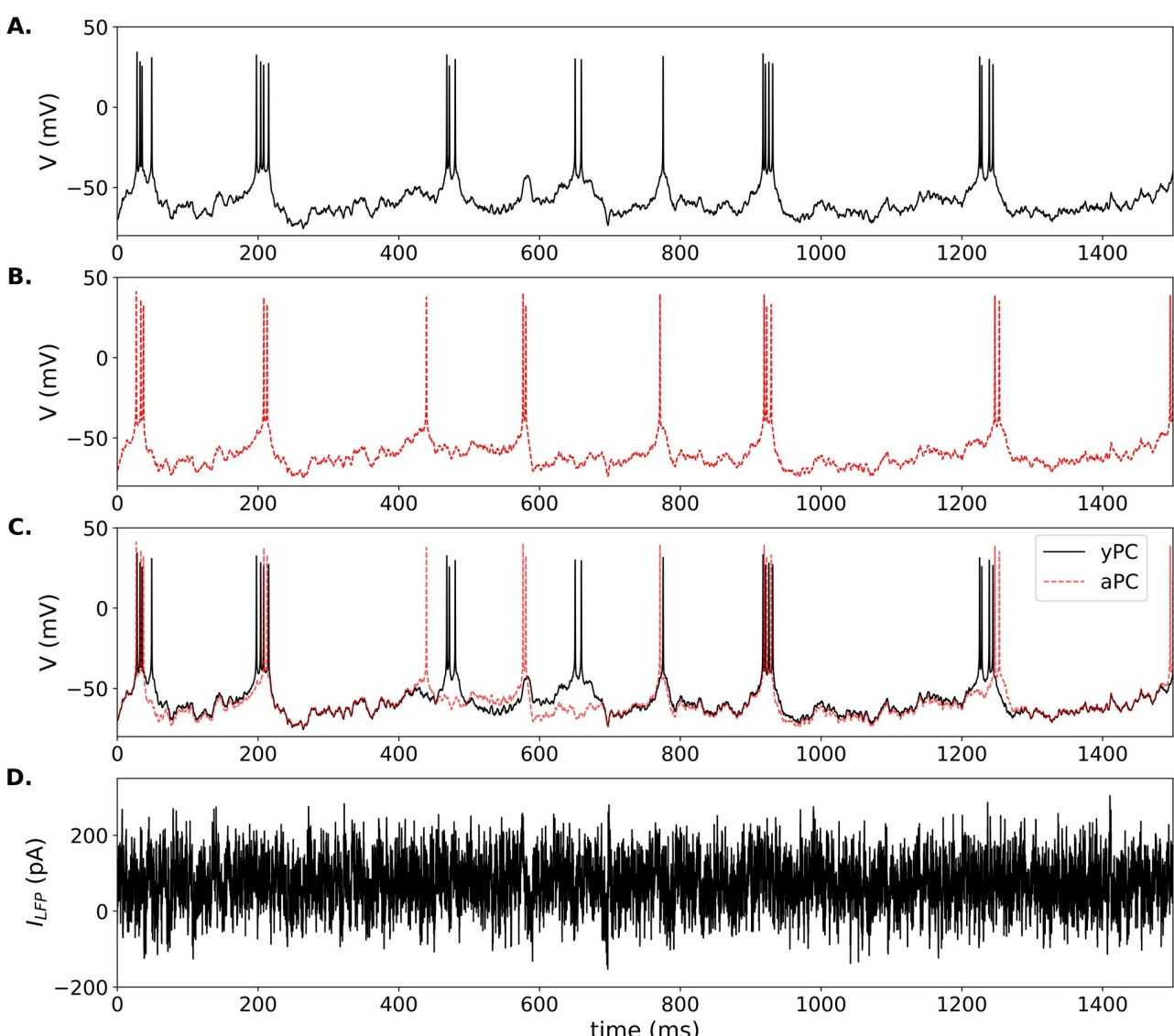

**Fig 6. LFP-induced bursting in model PCs.** Responses of the conditionally bursting yPC (solid black traces) and aPC (dashed red traces) models to the same LFP forcing. (A.) yPC response, (B.) aPC response, and (C.) overlap of the two traces. Parameters for the yPC and aPC are the same as in Fig 3. LFP parameters the same as in Fig 5 except $\sigma_F$ = 20.0 pA.

Alzheimer's in which altered cell firing could disrupt the balance [93]. We could also model the progression of aging in the network by varying the percentage of PCs which have altered $Ca^{2+}$ channel density, or implement a whole spectrum of channel expression across the simulated network. Studies like these may bring researchers closer to understanding the different stages of neurophysiological, or even pathophysiological, aging by allowing one to systematically vary biophysical parameters over a continuous range and identify points at which the system undergoes clear transitions in behavior.

## Aging and $Ca^{2+}$ channel expression

Our model can also reproduce changes in electrical activity seen in aged CA1 PCs, including larger AHPs [12–14] and increased adaptation [17–19], by increasing the L-type $Ca^{2+}$ current

amplitude to a level similar to that found in recordings. The L-type channel was modeled after the $Ca_v1.2$ isoform based on work showing this is the primary contributor in rodent brain, responsible for $\sim$70–80% of the L-type current [50, 94]. mRNA expression of *Cacnac1C* (the gene encoding $Ca_v1.2$) is increased in aged mice and rats [22, 95]. Increases in plasma membrane expression [25] and phosphorylation [96] of $Ca_v1.2$ channels have also been seen in aged rats. In addition, changes in *Cacna1c*/$Ca_v1.2$ expression are correlated with memory impairments [95, 97].

However, CA1 PCs also express the $Ca_v1.3$ isoform [98], which is responsible for $\sim$20% of the total L-type current [50, 94]. Studies have found both increased mRNA [22] and protein [24] expression of $Ca_v1.3$ in aged rats, and this increased expression is correlated with memory impairment [99]. Knockout studies in mice indicate that it is this isoform, and not $Ca_v1.2$, which contributes to slow AHP generation [100], possibly via activation of co-localized SK channels [98]. While experimental studies have been complicated by a lack of pharmacological agents which can isolate currents carried by the different isoforms, it would be relatively simple with our model to study the contributions of these two channels. The primary difference between the two is a shift in the activation curve of the $Ca_v1.3$ channel to more hyperpolarized values, relative to $Ca_v1.2$ [66]. Changing the parameter $v_j$ in Eq 7 would allow us to represent the different isoforms and explore how changes in the expression of each during aging might affect PC activity.

There are many cellular changes apart from $Ca^{2+}$ channel expression that occur during aging and could contribute to altered activity in PCs. Studies have implicated $Ca^{2+}$ release from intracellular stores as an important contributor, particularly to larger AHPs in aged animals (for reviews see [101, 102]). We did not explore the role of intracellular $Ca^{2+}$ stores in this study, nor many of the other cellular changes that surely contribute to the multifactorial process of aging. It is not our intention to suggest $Ca^{2+}$ channel expression is the only factor altering the activity of PCs in aged animals. Nevertheless, we do demonstrate that an increase in L-type $Ca^{2+}$ channels is *sufficient* to reproduce many of the changes in PC firing seen during aging. These results agree with a previous modeling study, which also found that an increase in L-type $Ca^{2+}$ conductance was sufficient to produce changes in adaptation and AHPs similar to those seen in aged CA1 PCs [103]. However, their model was mathematically complex, with 183 compartments and more than a dozen ionic currents—very different from the single-compartment, minimal model we present here. In addition, their study looked only at adaptation and AHPs, while our study goes further to investigate conditional bursting, spontaneous bursting, and responses to LFP stimulation.

## Aging and excitability

Our simulations do show changes in the electrical activity of aged PCs, but do these changes represent decreased excitability? This question relates more broadly to how we think about excitability—the term is rarely clearly defined or used in a standardized way. In some studies, excitability is used to refer to a change in the firing rate of a cell over the course of an injected current pulse, claiming that PCs with stronger adaptation are less excitable [18]. In our simulations under the adaptive firing parameter regime, the aPC did have stronger adaptation and fired fewer times during the stimulation period than the yPC. However, under some circumstances, this stronger adaptation occurred only after the aPC initially fired faster than the yPC (see for example Fig 2 inset). Should we consider this decreased excitability?

If what concerns us with excitability is the activity of the cell over a given time period, then the results under the bursting parameter regimes are even less clear. The aPC always fired fewer APs per burst than the yPC, indicating something akin to stronger adaptation. However,

if the 'event' we are considering is instead the burst, there are conditions under which the aPC fired a greater number of bursts in a given time period than the yPC. How should we interpret these results with respect to excitability? To our knowledge, there are very few experimental studies to date that have compared burst firing in young versus aged animal CA1 PCs, perhaps because of the relatively low percentage of cells with this firing pattern in certain developmental periods [83]. The only study of which we are aware compared spontaneous burst firing in young versus aged animal PCs during rest and exploratory behavior [104]. Unfortunately, there are several factors that make it difficult to compare our results with theirs. First, their recordings were extracellular. As such, no bursts from individual PCs are shown in the paper to compare to our simulations, which reproduce intracellular firing. Furthermore, the study focused on interspike intervals (ISIs), rather than the overall bursting pattern (e.g. number of bursts in a given period), which was our focus. Interestingly, Smith and colleagues [104] found that while ISIs were more left-skewed in aged animals when burst firing was recorded at rest, they found no difference between the age groups during behavior, suggesting that compensatory mechanisms work differently during behavior to adjust for changes in cellular excitability. Such compensatory mechanisms could be explored using our model, for example by adding simulated cholinergic input.

There are studies comparing bursting activity in young and aged animals in other areas of the brain. For example, Sagheddu and colleagues recorded from dopaminergic and GABAergic cells in the ventral tegmental area (VTA), and showed that both the percentage of spikes organized into bursts and the rate of bursting was lower in cognitively impaired aged rats [105]. In particular, their finding that the firing patterns of these cells in aged animals change from bursting to more single spiking (i.e. the percentage decrease) is consistent with some of our simulations.

Other researchers use the term excitability to describe how easy it is to get a cell to fire in response to stimulation, referring to "propensity" [106] or "readiness" [107]. In this context, excitability could be measured by the rheobase, or minimum current which generates firing in a neuron, as done in some studies of aging in CA1 [108]. However, 'propensity' or 'readiness' could also be interpreted as how quickly a cell fires after stimulus onset. In the AHP simulations, we saw that the aPC required additional current to fire the same number of spikes as for the yPC. On the other hand, the aPC often fired sooner than the yPC. These effects were a result of the increased $Ca^{2+}$ current in aged cells—the larger $Ca^{2+}$ current depolarized the cells faster and caused them to fire sooner, but it also caused the SK current to be larger and consequently slowed firing. It is as if the aPCs were initially more excitable, but then 'burned out' more quickly than the yPCs.

Overall, we believe our results highlight the importance of moving away from vague terms like 'excitability' in favor of precise language that describes the effect of interest (e.g., PCs spiked faster, spiked sooner, etc.). We hope that, in a broader sense, our study will encourage neuroscientists, and particularly aging researchers, to reevaluate how they think and write about excitability.

## Supporting information

**S1 Fig. Plots of model functions.** (A.) Forward ($\alpha_w$) and backward ($\beta_w$) rate functions, and the time constant ($\tau_w$), of activation of DK channels in the model. (B.) Steady-state activation curves for DK ($w_\infty$), $Na^+$($m_\infty$), and L-type $Ca^{2+}$($n_\infty$) channels. (C.) Full expressions and plots for each ion current in the model.
(TIF)

**S2 Fig. Adaptive firing in yPC.** (A.) Voltage response of the young model PC (yPC) in response to a 800 ms 150 pA square-pulse stimulation seen in (B.).
(TIF)

**S3 Fig. APs and currents underlying adaptive firing in yPC.** (A.) Two action potentials (APs) from the response seen in S2 Fig. (B.) Voltage- and $Ca^{2+}$-gated currents in the models as indicated in the legend, and their amplitudes and dynamics during the APs. Note that the $Na^+$-$K^+$pump current is not plotted due to its small amplitude.
(TIF)

**S4 Fig. Early and late adaptive firing in model PCs.** Further examination of the responses seen in Fig 1. (A.) Voltage responses of the yPC and aPC in the first $\sim 100$ ms of the square-pulse stimulation (6 vs. 4 spikes). (B.) Voltage responses of the yPC and aPC in the last $\sim 700$ ms of the square-pulse stimulation (4 vs. 2 spikes).
(TIF)

**S5 Fig. AHP in yPC.** (A.) Voltage response to a 100 ms square-pulse current injection of sufficient amplitude to elicit 4 APs in the yPC. Inset shows the amplitude and duration of the resulting afterhyperpolarization (AHP). (B.) Current pulse.
(TIF)

**S6 Fig. Conditional bursting in yPC.** (A.) Bursting in the yPC in response to a 800 ms 100 pA square-pulse current injection seen in (B.).
(TIF)

**S7 Fig. APs and currents underlying conditional burst firing in yPC.** (A.) Three APs from the response seen in S5 Fig. (B.) Voltage- and $Ca^{2+}$-gated currents in the models as indicated in the legend, and their amplitudes and dynamics during the APs. Note that the $Na^+$-$K^+$pump current is not plotted due to its small amplitude.
(TIF)

**S8 Fig. Conditional bursting in yPC and aPC compared at a single stimulation amplitude.** A. Bursting in the yPC (solid black traces) versus aPC (dashed red traces) in response to a 800 ms 100 pA square-pulse stimulation seen in (E.). Corresponding $Ca^{2+}$ currents, intracellular $Ca^{2+}$ concentration, and SK currents are shown in (B.), (C.), and (D.), respectively.
(TIF)

**S9 Fig. Spontaneous bursting in yPC.** (A.) Changes in membrane potential (i.e. spontaneous bursting) in the absence of current stimulation (B.).
(TIF)

**S10 Fig. APs and currents underlying spontaneous burst firing in yPC.** (A.) Three APs from the response seen in S8 Fig. (B.) Voltage- and $Ca^{2+}$-gated currents in the models as indicated in the legend, and their amplitudes and dynamics during the APs. Note that the $Na^+$-$K^+$pump current is not plotted due to its small amplitude.
(TIF)

**S11 Fig. Spontaneous bursting compared in yPC and aPC at single DK current amplitude.** A. Bursting in the yPC (solid black traces) versus aPC (dashed red traces) in the absence of stimulation (E.). Corresponding $Ca^{2+}$ currents, intracellular $Ca^{2+}$ concentration, and SK currents are shown in (B.), (C.), and (D.), respectively.
(TIF)

## Acknowledgments

We thank reviewers whose suggestions helped us to improve this manuscript.

## Author Contributions

**Conceptualization:** Erin C. McKiernan, Marco A. Herrera-Valdez, Diano F. Marrone.

**Formal analysis:** Erin C. McKiernan, Marco A. Herrera-Valdez.

**Funding acquisition:** Diano F. Marrone.

**Investigation:** Erin C. McKiernan, Marco A. Herrera-Valdez.

**Methodology:** Erin C. McKiernan, Marco A. Herrera-Valdez.

**Project administration:** Diano F. Marrone.

**Software:** Erin C. McKiernan, Marco A. Herrera-Valdez.

**Supervision:** Diano F. Marrone.

**Validation:** Erin C. McKiernan, Marco A. Herrera-Valdez.

**Visualization:** Erin C. McKiernan, Marco A. Herrera-Valdez.

**Writing – original draft:** Erin C. McKiernan, Marco A. Herrera-Valdez.

**Writing – review & editing:** Erin C. McKiernan, Marco A. Herrera-Valdez, Diano F. Marrone.

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
