## [Decision Letter · Decision Letter 0]

9 Jun 2024

PONE-D-24-14388A biophysical minimal model to investigate age-related changes in CA1 pyramidal cell electrical activityPLOS ONE

Dear Dr. McKiernan,

Thank you for submitting your manuscript to PLOS ONE. I apologize for the delay in reaching a decision. We managed to secure the opinion of one reviewer, who has assessed your manuscript very positively; their report is attached below. I would be happy to accept the paper for publication after it has been revised to address the points raised by the reviewer.

We look forward to receiving your revised manuscript.

Kind regards,

Jordi Garcia-Ojalvo

Academic Editor

PLOS ONE

Journal Requirements:

 [This work was supported by grants from the Natural Sciences and Engineering Research Council of Canada, as well as the Ontario Mental Health Foundation, awarded to DFM. This work was also supported by DGAPA-UNAM-PAPIIT IA209817 awarded to ECM; and by DGAPA-UNAM-PAPIIT IA208618 & IN228820, and DGAPA-UNAM-PAPIME PE114919 awarded to MAH-V.].  

Reviewers' comments:

Reviewer's Responses to Questions

**Comments to the Author**

1. Is the manuscript technically sound, and do the data support the conclusions?

Reviewer #1: Yes

2. Has the statistical analysis been performed appropriately and rigorously? 

Reviewer #1: N/A

3. Have the authors made all data underlying the findings in their manuscript fully available?

Reviewer #1: Yes

4. Is the manuscript presented in an intelligible fashion and written in standard English?

Reviewer #1: Yes

5. Review Comments to the Author

Reviewer #1: The model presented in this study is impressive, particularly given its simplicity: with just three variables (though many constants), with only one parameter to reproduce the age-related differences. It effectively reproduces the differences between young and old CA1 pyramidal cells in the hippocampus, such as adaptive firing, stimulus-induced bursting, and spontaneous bursting. I thoroughly enjoyed reading it and congratulate the authors on their impressive results, along with the well-done Jupyter notebook that complements the paper excellently. I encourage the authors to continue their work by exploring bifurcation analysis and conducting a more robust exploration of the model's parameter limitations compared to experimental values.

I hope my comments can help to improve this paper.

Major comments:

1) Abstract: The problem is well-introduced, and the rationale for the study is clear. However, please add some statements at the end about the conclusions of your research and its relevance to the field (what's the novelty?).

2) In the introduction you pose some questions [3-8] that are not fully answered by the paper. 2.1) I'd consider revisiting this questions in the discussion together with a mention on how this work could help in the field of AD or Parkinson's, or other age-related pathologies. 2.2) Can this be used to model the impairment of plastic mechanisms?

3) Methods [46-56]: Clarify better what distinguishes this model from the one previously developed.

4) Methods Section 2.1: Please provide a brief explanation of what I_{F} and I_{CaL} are when explaining Equation 1.

5) Clarification on s_x and N_x: The values of s_x are unclear for non-voltage-gated channels (NaT adn DK?), together with the values of N_x. Can you consider adding this information to the Table 1? Is there supporting literature from the values you choose?

6) Consider adding the figures of the Jupyter notebook in an Appendix/Supporting Material, so that it is easier to reference. They will help understanding the model functions and variables, contributions of I_x to V...

7) Recent paper of potential interest to add to the paper (bursting patterns aged vs. young, line 168-169): https://www.ncbi.nlm.nih.gov/pmc/articles/PMC10926450/ also consider mentioning it in the intro/discussion.

8) General comments to the figures: 8.1 There are no captions (?); 8.2 Please add the legend (yPC, aPC) + see next comment

9) Figure 1 (and methods 3.1).

9.1 (top panel) Not clear how do you compute the frequency (e.g., 60Hz vs. 40Hz). I think I’d be helpful to have in the figure a visual separation between the two segments you are referring to (first 100ms + the rest), plus a bar plot (for example) that shows quantitatively the differences between young and aged PCs.

9.2 Also, If you don’t want to include the figures in the notebook, at least add in the plot when the square pulse finishes and ends and with which amplitude.

9.3 Again, I’d add the rest of the figures (which are relevant in my opinion) in the appendix (at least).

9.4 Rephrase the text 197-201 so that it is easier to understand which panel you are referring to.

9.5 All above (9.1-9.3) also applies to the other figures (when applicable)!!

10) Parameter tuning: In the methods (and/or results), explain how parameters are tuned to find each firing pattern (I understand it is not only based on literature). Clarify which parameters are changed for each plot.

11) Figure 2, Bottom Panel: Indicate whether the graph shows saturation or continuous increase. If not changing the plot, mention in the caption that this is not a saturation effect.

12) Figures 5 and 6: Consider merging these figures as they contain overlapping information. Clearly indicate which parameters are changed between plots and the reasons for those changes.

Minor comments:

31 means → a means

46 CA1 in blue?

There is a problem with the references to the figures, and It was very hard to guess which figure you might be referring to.

Remove paragraph space between 290-291.

6. PLOS authors have the option to publish the peer review history of their article (what does this mean?). If published, this will include your full peer review and any attached files.

Reviewer #1: **Yes: **Roser Sanchez-Todo

---

## [Author Response · Author response to Decision Letter 0]

23 Jul 2024

We have detailed our responses to all formatting requests from the editor in our cover letter, and all the suggestions from the reviewer in our response to reviewers document included in our uploaded files. All changes can be seen in the tracked changes version of our manuscript.

---

## [Decision Letter · Decision Letter 1]

31 Jul 2024

A biophysical minimal model to investigate age-related changes in CA1 pyramidal cell electrical activity

PONE-D-24-14388R1

Dear Dr. McKiernan,

We’re pleased to inform you that your manuscript has been judged scientifically suitable for publication and will be formally accepted for publication once it meets all outstanding technical requirements.

Kind regards,

Jordi Garcia-Ojalvo

Academic Editor

PLOS ONE

**Comments to the Author**

1. If the authors have adequately addressed your comments raised in a previous round of review and you feel that this manuscript is now acceptable for publication, you may indicate that here to bypass the “Comments to the Author” section, enter your conflict of interest statement in the “Confidential to Editor” section, and submit your "Accept" recommendation.

Reviewer #1: All comments have been addressed

2. Is the manuscript technically sound, and do the data support the conclusions?

Reviewer #1: (No Response)

3. Has the statistical analysis been performed appropriately and rigorously? 

Reviewer #1: (No Response)

4. Have the authors made all data underlying the findings in their manuscript fully available?

Reviewer #1: (No Response)

5. Is the manuscript presented in an intelligible fashion and written in standard English?

Reviewer #1: (No Response)

6. Review Comments to the Author

Reviewer #1: (No Response)

7. PLOS authors have the option to publish the peer review history of their article (what does this mean?). If published, this will include your full peer review and any attached files.

Reviewer #1: **Yes: **Roser Sanchez-Todo

---

## [Editor Report · Acceptance letter]

9 Aug 2024

PONE-D-24-14388R1 

PLOS ONE

Dear Dr. McKiernan, 

I'm pleased to inform you that your manuscript has been deemed suitable for publication in PLOS ONE. Congratulations! Your manuscript is now being handed over to our production team.

Kind regards, 

on behalf of

Dr. Jordi Garcia-Ojalvo 

Academic Editor

PLOS ONE